# Predicting Teeth Extraction after Concurrent Chemoradiotherapy in Locally Advanced Nasopharyngeal Cancer Patients Using the Novel GLUCAR Index

**DOI:** 10.3390/diagnostics13233594

**Published:** 2023-12-04

**Authors:** Efsun Somay, Erkan Topkan, Busra Yilmaz, Ali Ayberk Besen, Hüseyin Mertsoylu, Ugur Selek

**Affiliations:** 1Department of Oral and Maxillofacial Surgery, Faculty of Dentistry, Baskent University, Ankara 06490, Turkey; efsuner@gmail.com; 2Department of Radiation Oncology, Faculty of Medicine, Baskent University, Adana 01120, Turkey; 3Department of Oral and Maxillofacial Radiology, School of Dental Medicine, Bahcesehir University, Istanbul 34349, Turkey; busra.yilmaz1@bau.edu.tr; 4Clinics of Medical Oncology, Adana Seyhan Medical Park Hospital, Adana 01120, Turkey; besenay@gmail.com; 5Clinics of Medical Oncology, Istinye University, Adana Medical Park Hospital, Adana 01120, Turkey; mertsoylu@hotmail.com; 6Department of Radiation Oncology, School of Medicine, Koc University, Istanbul 34450, Turkey; ugurselek@yahoo.com

**Keywords:** nasopharyngeal cancer, tooth extraction, glucose, C-reactive protein, albumin

## Abstract

To evaluate the value of the newly created GLUCAR index in predicting tooth extraction rates after concurrent chemoradiotherapy (C-CRT) in locally advanced nasopharyngeal carcinomas (LA-NPCs). **Methods:** A total of 187 LA-NPC patients who received C-CRT were retrospectively analyzed. The GLUCAR index was defined as ′GLUCAR = (Fasting **Glu**cose × **C**RP/**A**lbumin **R**atio) by utilizing measures of glucose, C-reactive protein (CRP), and albumin obtained on the first day of C-CRT. **Results:** The optimal GLUCAR cutoff was 31.8 (area under the curve: 78.1%; sensitivity: 70.5%; specificity: 70.7%, Youden: 0.412), dividing the study cohort into two groups: GLUCAR ˂ 1.8 (N = 78) and GLUCAR ≥ 31.8 (N = 109) groups. A comparison between the two groups found that the tooth extraction rate was significantly higher in the group with a GLUCAR ≥ 31.8 (84.4% vs. 47.4% for GLUCAR ˂ 31.8; odds ratio (OR):1.82; *p* < 0.001). In the univariate analysis, the mean mandibular dose ≥ 38.5 Gy group (76.5% vs. 54.9% for <38.5 Gy; OR: 1.45; *p* = 0.008), mandibular V55.2 Gy group ≥ 40.5% (80.3 vs. 63.5 for <40.5%, *p* = 0.004, OR; 1.30), and being diabetic (71.8% vs. 57.9% for nondiabetics; OR: 1.23; *p* = 0.007) appeared as the additional factors significantly associated with higher tooth extraction rates. All four characteristics remained independent predictors of higher tooth extraction rates after C-CRT in the multivariate analysis (*p* < 0.05 for each). **Conclusions:** The GLUCAR index, first introduced here, may serve as a robust new biomarker for predicting post-C-CRT tooth extraction rates and stratifying patients according to their tooth loss risk after treatment.

## 1. Introduction

Nasopharyngeal cancers (NPCs) rank as the 23rd most prevalent form of cancer globally, with an approximate total of almost 130,000 new cases reported annually [1]. They are highly aggressive malignant tumors that originate from the nasopharyngeal epithelium and significantly contribute to the morbidity and mortality associated with head and neck cancers. Despite significant progress in the field of diagnostic imaging and mass screening techniques, the majority of NPC patients (70–75%) present with locally advanced disease (LA-NPCs) due to the peculiar location of the malignancy [2,3]. Definitive platinum-based concurrent chemoradiotherapy (C-CRT) with intensity-modulated radiotherapy (IMRT) has superseded radiation alone or sequential chemoradiotherapy regimens in the treatment of medically fit LA-NPCs, as it has demonstrated a substantial improvement in locoregional disease control and survival rates, as well as a notable reduction in most severe toxicities [4,5].

In addition to its antitumor actions, high doses of ionizing radiation in the head and neck region may also harm healthy tissues within or near the radiation field. These tissues usually include the skin, muscles, oral mucosa, salivary glands, teeth, and upper and lower jaw bones, some of which are unavoidably encompassed by the planning target volume [6]. As a result, high doses of radiotherapy (RT) can cause a number of oral toxicities, including radiation caries, osteoradionecrosis, hyposalivation, dysgeusia, dysphagia, trismus, and tooth loss [7]. RT and C-CRT have been known to cause significant damage to teeth, vascularization, and supporting tissues, inevitably leading to tooth extractions and a decline in oral functions and related quality of life (QoL) measures [8]. Additionally, pre-C-CRT tooth extractions have been recently shown to be associated with weight loss > 5% during the C-CRT course in oropharyngeal cancer patients [9], a well-recognized predictor of poorer survival outcomes in almost all solid cancers [10]. Therefore, tooth loss before, during, or after oncological therapy may not only impair oral functioning but also contribute to malnutrition and a poor disease prognosis. From this perspective, it is essential to ascertain new biological indicators to precisely predict the likelihood of tooth loss at any point throughout cancer treatment, which might facilitate the timely implementation of preventative or therapeutic interventions for those at high risk.

Historically, little research has been conducted to investigate the impact of various biomarkers on the occurrence of tooth loss after C-CRT and RT. In a study conducted by Yilmaz et al. [11], it was demonstrated that among 263 patients with LA-NPC, those with low pretreatment hemoglobin (Hb) levels (Hb ≤ 10.6 g/dL) had a higher incidence of tooth extraction following C-CRT compared to those with high Hb levels (83.9% vs. 78.1% for Hb > 10.6 g/dL; *p* < 0.001). A separate investigation, including a cohort of 246 patients diagnosed with locally advanced squamous cell head and neck cancer, indicated a significant correlation between the need for tooth extraction after C-CRT and higher values of the systemic immune inflammation index (SII) measured before treatment initiation (*p* = 0.001). This inflammatory biomarker reflects the congruence between the patient’s inflammatory and immune status, regardless of the underlying cause, and was more common in the group with the determined SII cutoff value (cutoff: 558) compared to the group with lower values [12]. These studies have clarified that hypoxia-, immune-, and inflammation-related biomarkers can predict tooth loss after C-CRT for people with head and neck cancer, including LA-NPCs. Hence, further research in this field is now more promising than ever before.

Periodontal disease and dental caries are the most common causes of tooth loss, and inflammation plays a crucial role in both. Some inflammatory markers, such as high glucose levels in the blood, can cause microvascular and macrovascular changes that lead to periodontal disease and compulsory tooth extractions [13]. In a study conducted by Suzuki et al., using the National Database of Health Insurance Claims and Specific Health Checkups of Japan, it was shown that individuals belonging to the diabetes mellitus (DM) group had a greater prevalence of tooth loss compared to those in the control group, irrespective of gender [14]. Patients diagnosed with DM tended to lose their posterior teeth at earlier ages than those in the control group. Additionally, individuals within the DM cohort had a higher prevalence of tooth loss, irrespective of the presence or absence of periodontal disease treatment. Similarly, elevated levels of C-reactive protein (CRP), an established marker of acute and chronic inflammation, strongly indicate the presence of destructive periodontal disease and inevitable tooth loss [15,16,17,18]. Another acute-phase reactant protein that may be associated with tooth loss rates is albumin. Yoshihara et al. found a significant correlation between the number of missing teeth in 5 or 10 years and decreased serum albumin levels in patients with low serum albumin levels [19]. Likewise, the results of another study revealed that patients with hypoalbuminemia were at high risk for root caries and related tooth loss [20].

Based on the robust findings of the studies mentioned above, it can be confidently stated that elevated levels of glucose and CRP, along with reduced levels of albumin, serve as highly accurate predictors of tooth loss rates following hyperinflammatory conditions, including the LA-NPC patients undergoing definitive C-CRT. Therefore, motivated by the compelling evidence, we hypothesized that the integration of pretreatment glucose (GLU) and CRP-to-albumin ratio (CAR) measurements, namely, the GLUCAR index, should provide improved predictive capabilities for the unavoidable tooth extractions in patients with LA-NPC undergoing definitive C-CRT. Consequently, this retrospective research was conducted to examine the significance of the newly proposed GLUCAR index in predicting tooth loss rates after C-CRT in these patient groups.

## 2. Patients and Methods

### 2.1. Ethics, Consent, and Permission

The institutional review board of the Baskent University Medical Faculty approved the retrospective study design before compiling any data (project no: DKA:19/39). Eligible patients provided informed consent before undergoing oral and dental evaluations and C-CRT, allowing for collecting and analyzing blood samples, sociodemographic and medical data, dental X-rays, and academic presentations. This retrospective study was conducted in collaboration between the Department of Radiation Oncology and the Dentistry Clinics of the Baskent University Medical Faculty, and was approved by the institutional review board.

### 2.2. Patient Population

The Dentistry Clinics at Baskent University’s Adana Research and Treatment Center analyzed the records of LA-NPC patients who received C-CRT and had pre- and post-C-CRT oral and dental examinations between February 2010 and January 2023. To be included in this study, patients had to meet the following criteria: age of 18 years, histopathologic evidence of squamous cell carcinoma, locally advanced disease as per the 8th edition of the American Joint Cancer Committee (AJCC) cancer staging criteria, no previous history of other cancers, no history of systemic chemotherapy or head and neck RT, and accessible complete blood count and biochemistry test results before C-CRT. Access to pre- and post-C-CRT dental and panoramic radiography examination records was an absolute requirement for eligibility. Patients with tumor or lymph node invasion in the mandible, a previous diagnosis of osteoradionecrosis of jaws or a history of jaw surgery, and the use of steroids or other immune modifiers, as well as blood transfusions within 30 days before the start of C-CRT were all ineligible for the study. Patients with active systemic inflammatory diseases, such as rheumatological, nephritic, respiratory disorders, viral hepatitis, immune suppressive, collagen vascular, chronic inflammatory, and glucose storage diseases, were excluded from the study. These restrictions were deliberately implemented to reduce the possibility of biased effects resulting from pre-existing inflammatory and immunological diseases and medication usage. Furthermore, to mitigate the influence of their partiality on the results, individuals with periodontitis, cardiovascular diseases, vascular disorders, stroke, metabolic syndrome, and diabetes, which are among the variables that make individuals more susceptible to tooth loss, were also eliminated from this research.

### 2.3. Baseline Oral Examination

All patients received a comprehensive dental evaluation from a skilled oral and maxillofacial surgeon (ES) before C-CRT, following the guidelines of the American Dental Association (ADA) and the US Food and Drug Administration (FDA) [21]. Radiographic examinations were performed using panoramic scans as part of standard dental care for every head and neck cancer patient following the instructions provided by the manufacturer (J Morita, Veraviewepocs 2D, Kyoto, Japan). All teeth were examined for dental caries using World Health Organization criteria with illuminated and explorer mirrors [22]. Teeth that had no periodontal support, were too decayed to be restored, and had apical lesions that were too large to be treated with root canal treatment were extracted. Shallow tooth decay lesions were treated with the use of dental fillings. The patients received instruction on oral hygiene practices, and dental scaling procedures were conducted to promote the ongoing maintenance and enhancement of oral hygiene.

### 2.4. GLUCAR Index Calculation and Measurement

We developed the novel GLUCAR index as ***‘GLUCAR = [Fasting glucose (mg/dL) × CRP (mg/dL)/albumin (g/dL)]’*** by using pretreatment glucose, CRP, and albumin measures obtained from the standard complete blood count tests performed on the first day of the C-CRT. The Abbott Architect c8000 Biochemistry Autoanalyzer (Abbott Architect c8000 Biochemistry Autoanalyzer, Abbott, Chicago, IL, USA) was used for pretreatment measurements of fasting glucose, CRP, and albumin. The measurements were done following the manufacturer’s instructions [23].

### 2.5. Chemoradiotherapy Protocol

The RT technique used for all patients was simultaneous integrated boost intensity-modulated RT (SIB-IMRT), as previously described [11]. Coregistered imaging modalities, including computed tomography (CT), 18-fluorodeoxyglucose–positron emission tomography/CT (18-FDG-PET-CT), and magnetic resonance imaging (MRI), were used to delineate target volumes. The RT dosages were as presented earlier [12]: high-, intermediate-, and low-risk planning target volumes (PTV) received 70 Gy, 59.4 Gy, and 54 Gy, respectively, delivered in 33 daily fractions without treatment on weekends. Along with RT (every 21 days), three cycles of concurrent chemotherapy using cisplatin and 5-fluorouracil were advised. After C-CRT, all patients were advised to undergo two additional cycles of the identical chemotherapy protocol as adjuvant therapy. Antiemetics, dietary recommendations, and other supportive care were provided when necessary.

### 2.6. Follow-Up Oral Examination

The method described in the “Baseline oral examination” section was followed, and subsequent oral and dental examinations were conducted according to the scheduled timeline or as determined by clinical indications. Each patient’s clinical and radiological examination data were recorded at post-C-CRT 1, 3, 6, 9, and 12 months, and subsequently at every scheduled 6-month interval or whenever necessary. Based on the concepts highlighted in the “Baseline oral examination” section earlier, the treatment requirements for each patient were determined and reported.

### 2.7. Statistical Analysis

The primary endpoint was the connection between pretreatment GLUCAR index values and the requirement for tooth extractions during the post-C-CRT follow-up. The description of continuous variables included medians and ranges, while categorical variables were represented via percentage frequency distributions. Appropriate statistical analyses, such as the Chi-square test, Student’s *t*-test, or Spearman correlation, were used to compare the groups of patients. We used receiver operating characteristic (ROC) curve analysis to identify the pre-C-CRT cutoff(s) that could divide the entire research cohort into two groups with different outcomes. A logistic regression analysis was performed to identify the variables with multivariate significance. All comparisons were two-tailed, and a *p* ≤ 0.05 was considered significant.

## 3. Results

The current study examined 187 individuals diagnosed with LA-NPC who underwent C-CRT and fulfilled the necessary inclusion criteria. The baseline patient and disease characteristics of the entire study cohort are shown in Table 1. The age range for the total study was 18 to 78 years, with 56 years as the median age. The majority of study participants were male (67.4%), and the majority had T3–4 tumors (54.5%) and N2–3 nodal (63.6%) disease. In 67.9% and 57.8% of patients, respectively, there was a history of smoking or alcohol consumption. All patients underwent pre-C-CRT dental extractions, with a mean of 16 days (range: 10 to 22 days) between the tooth extractions and the initiation of C-CRT. Out of all the patients, 20.3% had diabetes. The median fasting glucose measure was 97 mg/dL (range: 71–194 mg/mL). The median CRP and albumin measures were 5.3 mg/dL (range: 0.4–39.4 mg/dL) and 37.2 g/dL (range: 23.4–51.7 g/dL), respectively.

The mean follow-up time was 48.3 months (range: 6–154.6 months). A total of 148 patients (79.1%) received two to three cycles of concurrent chemotherapy, while 136 patients (72.7%) received one to two cycles of adjuvant chemotherapy (Table 2). Tooth extraction was performed in 69.0% of patients during the follow-up period.

ROC curve analysis was used to determine fitting cutoff values for continuous variables, including the mean MMD, GLUCAR index, and Vx, for their potential interactions with post-C-CRT tooth extraction rates (Table 3). The GLUCAR index had an optimal cutoff point of 31.8 (AUC: 78.1%; sensitivity: 70.5%; specificity: 70.7%, Youden: 0.41), which divided the population into two groups with significantly different tooth extraction rates: the GLUCAR ˂ 31.8 (N = 78) and GLUCAR ≥ 31.8 (N = 109) groups (Figure 1). Based on the data presented in Table 1 and Table 2, it can be confidently stated that the distribution of pre-C-CRT characteristics and treatment features was almost identical between the two GLUCAR groups. However, a comparison between the two GLUCAR groups found that the rate of tooth extraction after C-CRT was significantly higher in patients with a GLUCAR ≥31.8 compared to those with a GLUCAR ˂ 31.8 (84.4% vs. 47.4%; OR: 1.82; *p* < 0.001). Additional ROC analysis was executed to explore the existence of any potential correlations between the mean mandibular dose (MMD), Vx, and tooth extraction rates. The results indicated that the ideal cutoffs were 38.5 Gy (AUC: 74.6%; sensitivity: 73.7%; specificity: 70.4%; Youden: 0.44) for MMD and 40.5% for V55.2 Gy (AUC: 80.7%; sensitivity: 78.2%; specificity: 75.6%; Youden: 0.54). Highlighting the strong correlations between the dosimetric parameters and tooth losses after C-CRT, tooth extraction rates were significantly higher in the MMD 38.5 Gy and V55.2 Gy ≥ 40.5% groups as compared to their MMD < 38.5 Gy and V55.2 Gy < 40.5% counterparts, respectively.

In the univariate analysis, a statistically significant association was observed between the rates of tooth extraction after C-CRT and the MMD ≥ 38.5 Gy group (76.5% vs. 54.9% for <38.5 Gy, *p* = 0.008, OR: 1.45), the mandibular V55.2 Gy ≥ 40.5% group (80.3% vs. 63.5% for <40.5%, *p* = 0.004, OR: 1.30), and the presence of DM (71.8% vs. 57.9% for absent, *p* = 0.007, OR: 1.23). As shown in Table 3, the results of multivariate analyses indicated that each of the four characteristics examined remained significant and independent predictors of higher tooth extraction rates in patients with LA-NPC treated with definitive C-CRT (*p* < 0.05 for each) (Figure 2).

## 4. Discussion

The major objective of this study was to determine whether the incidence post-C-CRT-TE rates in patients with LA-NPC who had definitive C-CRT could be predicted using the GLUCAR index. In this research, we made a first attempt to investigate the association in this patient population. The findings of our study showed that the novel pretreatment GLUCAR index was successful in dividing LA-NPC patients into two groups based on their likelihood of requiring tooth extractions following C-CRT (*p* < 0.001).

The most significant finding of our investigation was the discovery of the pre-C-CRT GLUCAR index as a novel and efficacious biomarker for predicting tooth extractions after C-CRT in patients with LA-NPC. Accordingly, the incidence of C-CRT tooth extraction rates was significantly higher in GLUCAR ≥ 31.8 patients than in their GLUCAR < 31.8 counterparts (84.4 vs. 47.4%, OR: 1.82; *p* < 0.001). This finding suggests that the GLUCAR index can divide such patients into two risk groups for post-C-CRT tooth extractions, which may guide the prompt implementation of preventive measures in patients with a higher risk of tooth loss. Although it is a challenging task to discuss these first-of-its-kind results comparatively, studies on the effects of the components of this unique index may help form a rational scientific basis for them. Elevated glucose levels are correlated with persistent chronic infection, inflammation, periodontitis, and increased radiosensitivity in most tissues, including the mandible. Moreover, hyperglycemia leads to the overproduction of glycoproteins that coat the epithelial linings and have the potential to cause thickening and obstruction of blood vessels [24]. These factors contribute to compromised immune responses and delayed healing mechanisms in the affected tooth apex, root, and alveolar ligaments, ultimately resulting in tooth loss [25]. Hyperglycemic patients are at a higher risk of developing dental caries and experiencing tooth loss, even in the absence of RT, because of the reduced salivary flow rates and elevated glucose levels in the saliva, which are expected to be further exacerbated when RT is present [26,27]. In confirmation, a recent meta-analysis conducted by Weijdijk and colleagues confirmed that patients with DM have a 1.63 times (*p* < 0.00001) higher risk of tooth loss compared to individuals without diabetes [28]. Similarly, according to Kuo et al., patients with HNC and DM who received C-CRT were more susceptible to infection, hematological toxicity, weight loss, and treatment-related mortality than those without DM, indicating that DM causes hypersensitivity to RT [29]. Above findings are also supported by the observation of significantly higher tooth extractions in the DM group than their nondiabetic counterparts (71.8% vs. 57.9%; OR: 1.23; *p* = 0.007).

The second and third components of the GLUCAR index are CRP and albumin, which can be examined separately or as a combination of them, namely, the CRP-to-albumin ratio (CRP/albumin). The anabolic process of CRP is heightened under inflammatory conditions, in contrast to the catabolic breakdown of albumin. As a result, there is a well-established and robust inverse correlation between the levels of CRP and albumin. Indeed, it has been observed that the liver responds promptly to a single inflammatory stimulus by synthesizing CRP and causing a rapid increase in its levels [30]. However, the reactionary release of TNF-α and IL-6 in response to this stimulus leads to a decrease in serum albumin levels, which can be attributed to an increase in the breakdown of albumin and a reduction in its synthesis in the liver [31]. Previous investigations have revealed that CRP plays a vital role in the development of periodontitis, a significant cause of tooth loss, by causing the production of inflammatory mediators, such as IL-1, IL-6, and TNF-α [32,33,34,35]. Considering the albumin, previously, two studies by Yoshihara et al. established a significant connection between hypoalbuminemia and the number of missing teeth in 5 or 10 years [19], and the risk of root caries and related tooth loss [20]. Furthermore, Ando et al. [36] conducted a study using data from a large-scale community-based Japanese population, revealing a significant correlation between decreased albumin levels and an elevated likelihood of experiencing tooth loss. Low albumin levels that signify malnutrition and exacerbated inflammation may be the root cause of tooth extraction, as these two factors may adversely impact the teeth and supporting tissues, ultimately leading to the need for extraction [37,38]. The study conducted by Keinänen et al. investigated the influence of an elevated CRP/albumin ratio on patients. The findings revealed that those who had tooth loss exhibited a notably higher CRP/albumin ratio compared to those without tooth loss [39]. Therefore, although the exact mechanism may be more complex, available research findings and those presented here cumulatively suggest that the high GLUCAR levels may indicate a poorer inflammatory, immune, and nutritional status, which may lead to higher tooth loss rates after C-CRT in LA-NPC patients.

Another vital implication of our findings may be that a high pretreatment GLUCAR index may behave either as an early sign of occult tooth, periodontal tissue, or jaw bone pathologies, or a surrogate marker of radiation or chemoradiation hypersensitivity of these tissues in such patients. This remark is supported by an earlier study investigating the utility of pretreatment SII, another immune–inflammation marker, in predicting teeth caries and the need for tooth extraction after C-CRT in locally advanced HNC patients [12]. The researchers of this study discovered a positive correlation between a high pretreatment SII measure and a statistically significant rise in the post-treatment tooth extraction rates group (77.1% for SII > 558 vs. 51.4% for SII ≤ 558; r_s_: 0.89; HR: 1.68; *p* = 0.001). In this particular context, it is possible that future research endeavors will validate these findings by performing correlative analyses between the radiomic and proteomic characteristics and immune–inflammation indices. This approach has the potential to improve the predictive capability of the results, particularly if it leads to the development of novel nomograms that exhibit high levels of predictive sensitivity and specificity.

The present investigation identified two additional parameters significantly associated with tooth extraction rates after C-CRT. These factors were MMD ≥ 38.5 Gy (76.5% vs. 54.9% for <38.5 Gy; OR: 1.45; *p* = 0.008) and mandibular V55.2 Gy ≥ 40.5% (80.3 vs. 63.5 for <40.5%; OR: 1.30; *p* = 0.004). Although the cutoff values differ, these results are well concordant with previous research suggesting higher tooth loss rates with increasing MMD and mandibular Vx values. The study carried out by Gomes-Silva et al. [40] examined the correlation between three distinct RT dosage levels (<30 Gy, 30–60 Gy, and >60 Gy) and the occurrence of tooth extraction after RT in patients with HNC. The likelihood of tooth extraction was found to be nearly three times higher with doses > 60 Gy compared to doses < 60 Gy (*p* < 0.001). Furthermore, an earlier investigation showed that the first degradation of the dental hard tissues became apparent even at doses between 30 and 60 Gy [41]. Research has indicated that the higher demand for tooth extraction following high RT doses can be attributed to several factors. RT leads to the development of porous areas in the enamel, resulting in a significant loss of the surface enamel and the protective coating it provides, which exposes the underlying dentin and impairs the natural remineralization process. Additionally, teeth may lose their resistance to acids and bacteria, as RT reduces the capacity of saliva to buffer acids, decreases the presence of antimicrobial antibodies and proteins, and diminishes salivary flow [40,41,42]. In addition, a study reported that most of the tooth loss occurring in the post-RT period occurred due to periodontal disease, apical periodontitis, and radiation caries, where high doses of RT may facilitate the development and progression of all these conditions [40]. While acknowledging the lack of a comparable study, the existing indirect evidence substantiates the findings of our investigation. In the research conducted by Yilmaz et al., it was observed that, out of 210 patients with LA-NPC, 174 individuals (84.8%) who underwent an MMD > 44.2 Gy had at least one tooth extraction during the post-C-CRT interval [43]. Similarly, a separate study demonstrated that 209 (79.5%) of 263 patients who underwent an MMD > 36.2 Gy and a V59.8 Gy > 36% reported tooth extraction in the post-C-CRT period [11]. Therefore, based on the findings of various studies, it is advisable to focus on minimizing the RT doses that affect the teeth and their surrounding structures, which can help reduce the likelihood of tooth loss after treatment, ultimately leading to an improved QoL for such patients.

The current investigation faces several limitations. First, the research relied on data obtained from retrospective analyses conducted inside a single institution with a relatively small study population. This characteristic introduces the possibility of inadvertent selection biases often associated with studies of this kind. Second, the lack of a validation cohort may have constrained our capacity to provide more robust interpretations of our findings. This limitation mainly stems from the restricted sample size of patients included in our study. Third, because our research only considered measurements taken at a single time point, the first day of C-CRT, it is crucial to assume that the existing cutoff values of GLUCAR may not accurately represent the optimal threshold for the best-fit risk stratification of LA-NPC patients. This is due to the potential for dramatic fluctuations of the blood albumin, CRP, and glucose levels during and after C-CRT. And fourth, we may have missed the opportunity to ascribe any plausible mechanistic associations between a cohort exhibiting an elevated GLUCAR score and the levels of cytokines/chemokines, nutritional status, and immune–inflammatory factors, such as IL-1, IL-6, and TNF-α. As a result, it is important to consider that the conclusions presented in this study should be seen as hypothetical rather than definitive recommendations until more well-designed large-scale research studies addressing these concerns provide supporting evidence. Nevertheless, the newly developed GLUCAR index components are easy to acquire and calculate, are affordable, and have reproducible factors, rendering it a suitable candidate biomarker for routine clinical use. Hence, despite the abovementioned constraints, the recently developed GLUCAR index can categorize LA-NPC patients into high- and low-risk groups based on their probability of experiencing tooth loss after C-CRT. Therefore, if confirmed with further research, its routine use may lead to close monitoring of high-risk patients and prompt implementation of proactive measures to avoid tooth loss at earlier stages.

## 5. Conclusions

The present research results demonstrate that the newly developed GLUCAR index is a dependable biomarker that successfully predicts the occurrence rates of post-C-CRT tooth extractions in patients with LA-NPC. This novel biological indicator may represent a significant milestone in identifying high-risk patients and may pave the way for the design of potent preventive measures and follow-up algorithms if further studies validate the results presented here.

## Figures and Tables

**Figure 1 diagnostics-13-03594-f001:**
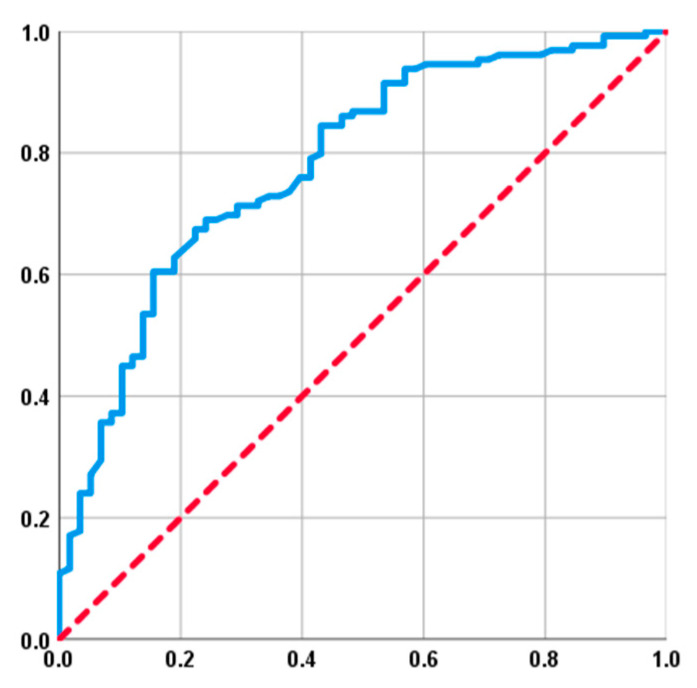
The results of a receiver operating characteristic (ROC) curve analysis investigating the relationship between pretreatment GLUCAR index values and tooth extraction rates after concurrent chemoradiotherapy (AUC: 78.1%; sensitivity: 70.5%; specificity: 70.7%, Youden: 0.41). Bule line represents Receiver Operating Curve; Red line represents no discrimination line.

**Figure 2 diagnostics-13-03594-f002:**
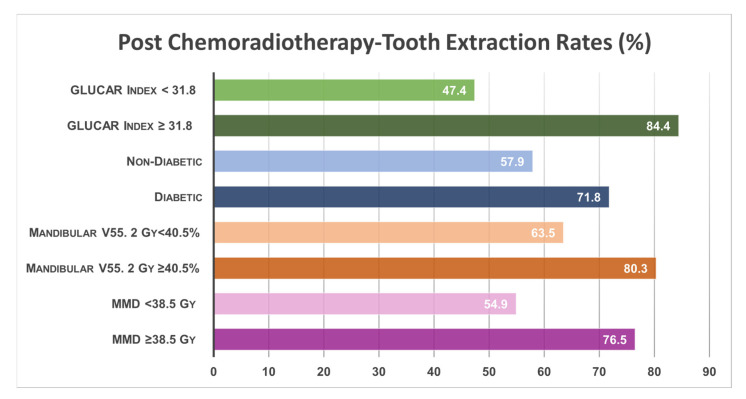
The bar chart showing post-C-CRT tooth extraction rates per factor demonstrated independent significance in multivariate analyses. Abbreviations: V, volume; Gy, gray; MMD, mean mandibular dose.

**Table 1 diagnostics-13-03594-t001:** Baseline and treatment characteristics for the entire study cohort and per GLUCAR index groups.

Characteristics	All Patients (N = 187)	GLUCAR Index ˂ 31.8 (N = 78)	GLUCAR Index ≥ 31.8 (N = 109)	*p* Value
Median age, years (range)	56 (18–78)	61 (18–77)	54 (18–78)	0.10
Age group				
≥56	99 (52.9)	30 (38.5)	69 (63.3)	0.001
<56	88 (47.1)	48 (61.5)	40 (36.7)	
Gender, N (%)				
Female	61 (32.6)	29 (37.2)	32 (29.4)	0.23
Male	126 (67.4)	49 (62.8)	77 (70.6)	
Smoking status, N (%)				
Yes	127 (67.9)	49 (62.8)	78 (71.6)	0.27
No	60 (32.1)	29 (37.2)	31 (28.4)	
Alcohol consumption, N (%)				
Yes	79 (42.2)	32 (53.8)	47 (43.1)	0.88
No	108 (57.8)	46 (46.2)	62 (56.9)	
Median number of pre-CCRT tooth extraction, N (%), (range)	4 (1–11)	4 (1–11)	4 (1–9)	0.3
Median tooth extraction time to C-CRT, days (range)	16 (10–22)	16 (10–22)	16 (12–22)	0.41
T-stage group, N (%)				
1–2	85 (45.5)	42 (53.8)	43 (39.4)	0.06
3–4	102 (54.5)	36 (46.2)	66 (60.6)	
N-stage group, N (%)				
0–1	68 (36.4)	31 (39.7)	37 (33.9)	0.44
2–3	119 (63.6)	47 (60.3)	72 (66.1)	
Median fasting glucose, mg/dL	97 (71–194)	93 (71–156)	104 (85–194)	0.03
Diabetes mellitus status, N (%)				
Yes	38 (20.3)	11 (14.1)	27 (24.8)	0.01
No	149 (79.7)	67 (85.9)	82 (75.2)	
Median CRP, mg/dL	5.3 (0.4–39.4)	3.4 (0.4–26.4)	6.1 (2.6–39.4)	0.002
Albumin, g/dL	37.2 (23.4–51.7)	41.7 (24.1–50.6)	32.2 (23.4–51.7)	0.008

Abbreviations: C-CRT, concurrent chemoradiotherapy; T, tumor; N, node; mg, milligram; dL, deciliter; g, gram.

**Table 2 diagnostics-13-03594-t002:** The connection between postconcurrent chemoradiotherapy characteristics of the whole study group and two GLUCAR index groups.

Characteristics	All Patients (N = 187)	GLUCAR Index ˂ 31.8 (N = 78)	GLUCAR Index ≥ 31.8 (N = 109)	*p*-Value
Concurrent chemoradiotherapy cycles, N (%)				
1	39 (20.9)	13 (16.7)	26 (23.9)	0.18
2–3	148 (79.1)	65 (83.3)	83 (76.1)	
Adjuvant chemoradiotherapy cycles, N (%)				
0	51(27.3)	19 (24.4)	32 (29.3)	029
1–2	136 (72.7)	59 (75.6)	77 (707)	
Median MMPD; Gy (range)	46.8 (30.4–73.4)	46.3 (31.7–71.6)	47.4 (30.4–73.4)	0.51
Post-CCRT tooth extraction, N (%)				
Absent	58 (31.0)	41 (52.6)	17 (15.9)	˂0.001
Present	129 (69.0)	37 (47.4)	92 (84.1)	
Median post-C-CRT extracted tooth, N (range)	1 (0–6)	0 (0–4)	1 (0–6)	˂0.001
Median time from C-CRT to tooth extraction, mo. (range)	7 (2–19)	6 (5–18)	9 (2–19)	0.003
Time of post-CCRT tooth extraction, N (%) *				
≤6	68 (52.7)	19 (51.4)	49 (53.3)	0.50
>6	61 (47.3)	18 (48.7)	43 (46.7)	
MMD, Gy (range)	33.2 (10.1–50.4)	33.9 (10.7–50.4)	32.6 (10.1–50.1)	0.39
MMD group, N (%)				
<38.5 Gy	102 (54.5)	43 (55.1)	59 (54.1)	0.72
>38.5 Gy	85 (45.5)	35 (44.9)	50 (45.9)	
V55.2 Gy group, N (%)				
<40.5%	126 (67.4)	53 (67.9)	73 (67.0)	0.69
≥40.5%	61 (32.6)	25 (32.1)	36 (33.0)	

Abbreviations: MMPD, median maximum mandibular point dose; Gy, gray; C-CRT, concurrent chemoradiotherapy; MMD, mean mandibular dose; V, volume. Note: * For 129 patients who underwent post-C-CRT tooth extraction (for the GLUCAR index ˂ 31.8 group (N = 37) and the GLUCAR index ≥31.8 group (N = 92)).

**Table 3 diagnostics-13-03594-t003:** The univariate and multivariate results.

Factors	Post-C-CRT TE (%)	Univariate *p*-Value	Multivariate *p*-Value	OR
Age group (≥56 y vs. <56 y)	68.2 vs. 69.7	0.86	-	0.97 (0.91–1.15)
Gender (female vs. male)	67.7 vs. 73.0	0.94	-	0.83 (0.67–1.42)
Smoking status (yes vs. no)	70.1 vs. 66.7	0.74	-	1.06 (0.82–1.28)
Alcohol consumption (yes vs. no)	71.2 vs. 65.6	0.27	-	1.09 (0.92–1.23)
T-stage group (3–4 vs. 1–2)	70.4 vs. 67.8	0.33	-	1.11 (0.84–1.37)
N-stage group (2–3 vs. 0–1)	74.3 vs. 64.6	0.20	-	1.15 (0.94–1.38)
Concurrent chemotherapy cycles (2–3 vs.1)	70.3 vs. 59.4	0.026	0.084	1.28 (0.97–1.96)
Adjuvant chemotherapy cycles (1–2 vs. 0)	72.1 vs. 60.8	0.16	-	1.16 (0.93–1.37)
MMD group (≥38.5 Gy vs. <38.5 Gy)	76.5 vs. 54.9	0.008	0.014	1.45 (1.20–1.96)
Mandibular V55. 2 Gy group (≥40.5% vs. <40.5%)	80.3 vs. 63.5	0.004	<0.001	1.30 (1.10–1.6)
Diabetes mellitus group (present vs. absent)	71.8 vs.57.9	0.007	0.007	1.23 (1.12–1.4)
GLUCAR index group (≥31.8 vs. ˂31.8)	84.4 vs. 47.4	˂0.001	<0.001	1.82 (1.47–2.33)

Abbreviations: C-CRT, concurrent chemoradiotherapy; TE, tooth extraction; y, year; T, tumor; N, node; MMD, mean mandibular dose; Gy, gray; V, volume; OR, odds ratio.

## Data Availability

The datasets used and/or examined in the present work are available to qualified researchers who meet the requirements for accessing sensitive data. These datasets may be obtained from the Baskent University Department of Radiation Oncology Institutional Data Access by contacting adanabaskent@baskent.edu.tr.

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
