# Peer review of "Predicting Teeth Extraction after Concurrent Chemoradiotherapy in Locally Advanced Nasopharyngeal Cancer Patients Using the Novel GLUCAR Index"

_diagnostics, 2023, doi:10.3390/diagnostics13233594_

Round 1

Reviewer 1 Report

Comments and Suggestions for Authors

1. While choosing a patient population for human study, have you considered other pathological reasons for tooth loss and ruled out the patients with those confounding factors?

2. Were glucose, CRP, and albumin measured by the standard and widely used quantitative approaches? Please include the necessary references as a demonstration. 

3. As you mentioned on line 170, multiple statistical methodologies are available to analyze the correlation between the indicators and the phenotype. What is the reason that a parameter test, t-test, was used in the study, rather than a non-parameter test? Chi-square and log-rank tests are also widely utilized in the difference and survival analysis in the biomedical field. 

Author Response

Reviewer 1.

We thank Reviewer 1 for his/her precious suggestions

Comment 1. While choosing a patient population for human study, have you considered other pathological reasons for tooth loss and ruled out the patients with those confounding factors?

Response 1.  This research also excluded patients who had periodontitis, cardiovascular disorders, vascular diseases, stroke, metabolic syndrome, and diabetes mellitus since these conditions are known to increase the risk of tooth loss. The exclusion criteria section has been revised to provide clarity on this matter.

Comment 2. Were glucose, CRP, and albumin measured by the standard and widely used quantitative approaches? Please include the necessary references as a demonstration. 

Response 2. The Abbott Architect c8000 Biochemistry Autoanalyzer (Abbott Architect c8000 Biochemistry Autoanalyzer, Abbott, USA) was used for pretreatment measurements of fasting glucose, CRP, and albumin. The measurements were done following the manufacturer's instructions (Pauli D, Seyfarth M, Dibbelt L. The Abbott Architect c8000: analytical performance and productivity characteristics of a new analyzer applied to general chemistry testing. Clin Lab. 2005;51(1-2):31-41. PMID: 15719702.). This information has been included in the section titled "GLUCAR Index Calculation and Measurement" and cited in the References section.

Comment 3. As you mentioned on line 170, multiple statistical methodologies are available to analyze the correlation between the indicators and the phenotype. What is the reason that a parameter test, t-test, was used in the study, rather than a non-parameter test? Chi-square and log-rank tests are also widely utilized in the difference and survival analysis in the biomedical field. 

Response 3. The decision to use a t-test was based on the almost normal distribution of numerical values. However, upon revision of our analyses, as recommended, we found that the p-values remained essentially unchanged and did not significantly impact the outcomes. Hence, we preserved the original p-values in the revised manuscript. The logistic regression analysis was intentionally chosen for the study as it does not show dependence on the time factor. However, it may also be appropriate to perform a Cox regression analysis if the time factor is accounted for. Hence, either of the statistical methods used here or recommended by the reviewer are valid for the study's purpose.

Reviewer 2 Report

Comments and Suggestions for Authors

Predicting Teeth Extraction After Concurrent Chemoradiother- 2 apy in Locally Advanced Nasopharyngeal Cancer Patients Using 3 the Novel GLUCAR Index.

The paper has good quality and recommend for publication after minor revision.

Nasopharyngeal cancer is the keywords of manuscript, but completely missed in the main text. 

Nasopharyngeal cancer must be discussed in the main text with its treatment approaches

Please up-date old references (1,2,3, 4, and ect).

The novelty of the study must be highlighted in the abstract and conclusion sections.

Please extend the figure.2 (Post-C-CRT-TE Rates (%)) caption’s.

I recommend the highlighting of advantages and disadvantages of “Novel GLUCAR Index” 

Comments on the Quality of English Language

Minor editing of English language required

Author Response

Reviewer 2

We express our gratitude to Reviewer 2 for their invaluable and insightful critique, which will contribute significantly to improving the quality of our research.

Comment 1-2. Nasopharyngeal cancer is the keywords of manuscript, but completely missed in the main text.  Nasopharyngeal cancer must be discussed in the main text with its treatment approaches

Response 1-2. We added a new paragraph to the beginning of the Introduction section, which discusses the specified recommendations and includes relevant references.

Comment 3. Please up-date old references (1,2,3, 4, and ect).

Response 3. All references in the manuscript were reviewed and updated as necessary.

Comment 4. The novelty of the study must be highlighted in the abstract and conclusion sections.

Response 4. The study's novelty has been emphasized in the manuscript's Abstract and Conclusion sections, as recommended.

Comment 5. Please extend the figure.2 (Post-C-CRT-TE Rates (%)) caption’s.

Response 5. The caption for Figure 2 has been updated.

Comment 6. I recommend the highlighting of advantages and disadvantages of the “Novel GLUCAR Index” 

Response 6. The revised manuscript discusses the advantages and disadvantages of the GLUCAR index in the Limitations section, as recommended.